# Defining the Role of the miR-145—KLF4—αSMA Axis in Mitral Valvular Interstitial Cell Activation in Myxomatous Mitral Valve Prolapse Using the Canine Model

**DOI:** 10.3390/ijms25031468

**Published:** 2024-01-25

**Authors:** Vicky K. Yang, Nicole Moyer, Runzi Zhou, Sally Z. Carnevale, Dawn M. Meola, Sally R. Robinson, Guoping Li, Saumya Das

**Affiliations:** 1Department of Clinical Science, Cummings School of Veterinary Medicine, Tufts University, North Grafton, MA 01536, USAsally.robinson@tufts.edu (S.R.R.); 2Cardiovascular Research Center, Massachusetts General Hospital, Boston, MA 02114, USA

**Keywords:** mitral valve prolapse, animal model, non-coding RNA, extracellular vesicles, translational model

## Abstract

Mitral valve prolapse (MVP) is a common valvular disease, affecting 2–3% of the adult human population and is a degenerative condition. A total of 5–10% of the afflicted will develop severe mitral regurgitation, cardiac dysfunction, congestive heart failure, and sudden cardiac death. Naturally occurring myxomatous MVP in dogs closely resembles MVP in humans structurally, and functional consequences are similar. In both species, valvular interstitial cells (VICs) in affected valves exhibit phenotype consistent with activated myofibroblasts with increased alpha-smooth muscle actin (αSMA) expression. Using VICs collected from normal and MVP-affected valves of dogs, we analyzed the miRNA expression profile of the cells and their associated small extracellular vesicles (sEV) using RNA sequencing to understand the role of non-coding RNAs and sEV in MVP pathogenesis. *miR-145* was shown to be upregulated in both the affected VICs and sEV, and overexpression of *miR-145* by mimic transfection in quiescent VIC recapitulates the activated myofibroblastic phenotype. Concurrently, KLF4 expression was noted to be suppressed by *miR-145*, confirming the *miR-145*—KLF4—αSMA axis. Targeting this axis may serve as a potential therapy in controlling pathologic abnormalities found in MVP valves.

## 1. Introduction

Mitral valve prolapse (MVP) is a common valvular disease, affecting 2–3% of the adult human population [1]. MVP can arise with a syndromic condition like Marfan’s syndrome, but it more commonly manifests as a non-syndromic condition [2], such as myxomatous mitral valve prolapse. While many patients remain asymptomatic, myxomatous MVP is a progressive and degenerative condition, and 5–10% of the afflicted will develop severe mitral regurgitation, cardiac dysfunction, congestive heart failure, and sudden cardiac death [1,2,3]. Currently, medical therapy for MVP is limited, and mitral valve repair or replacement is most effective when performed before the onset of congestive heart failure [4]. One obstacle to developing early intervention and medical therapy is the lack of rodent models that accurately recapitulate the disease evolution, as MVP in humans is polygenic heritable disease [5,6,7]. A robust animal model will aid in the understanding of the molecular drivers for MVP. Naturally occurring myxomatous MVP in dogs closely resembles myxomatous MVP in humans structurally, and functional consequences are similar, including congestive heart failure at the end stage of the disease. MVP is the most common acquired cardiac disease and the most common cause of congestive heart failure in dogs, comprising 2/3 of all canine cardiac cases [8]. This disease is also age-related, with a lifetime prevalence in older dogs (>9 years) over 90% [9].

The common cellular-molecular pathology in myxomatous MVP seen in humans and dogs is activation of the quiescent valve interstitial cells (VICs) (low alpha-smooth actin expression (α-SMA^low+^)) to activated myofibroblasts (α-SMA^high+^) and dysregulated extracellular matrix, including increased amount of glycosaminoglycan and proteoglycans within the valve leaflets, while the native collagen and elastin fibers are fragmented [10,11,12,13,14]. The exact mechanism that leads to myofibroblastic transformation is unknown, although both serotonin and transforming growth factor-beta (TGF-β) stimulation have been implicated [15,16], and dysregulation of non-coding RNAs such as micro (mi)RNA has been associated with MVP. We have demonstrated that cultured VIC harvested from MVP-affected canine valves have altered cellular miRNA expression compared to those from normal mitral valves (downregulation of *let-7c*, *miR-17*, *miR-20a*, and *miR-30d*) [14]. Cellular phenotype can also be influenced by genetic, cytokine, and protein signals contained within exogenous extracellular vesicles [17]. Small extracellular vesicles (sEV) such as exosomes are nanoscale vesicles ranging in size from 40 to 120 nm [17]. The genomic content, including miRNA, of sEV is highly regulated by the cell of origin and determined by the pathological state at the time of sEV production [18]. We have shown that changes in plasma sEV-associated miRNA (*miR-9, miR-181c*, *miR-495*, and *miR-599*) are detected in dogs with MVP and congestive heart failure [19]. Changes in circulating miRNA may be both manifestations of systemic alterations, including the heart or other organs whose functions are affected by cardiac dysfunction, as well as a reflection of the changes in cellular transcriptomes of the organ of interest (e.g., heart). As both cellular and sEV-associated miRNA play an important role in determining cellular phenotype, we hypothesized that VIC cellular and sEV-associated miRNA expression is dysregulated with MVP, and that dysregulated expression of certain miRNA species are shared between the activated myofibroblastic VICs and their associated sEVs. We confirmed that *miR-145* is upregulated in both MVP-associated VICs and sEV, and we further investigated the functional effects of *miR-145* in VICs. Our data showed that *miR-145*, by directly blocking the protein expression of KLF4, promotes the transition of quiescent VICs to a myofibroblastic phenotype, resulting in an increase in α-SMA expression. Our results showed that *miR-145* may be a key regulator in VIC phenotypic transition that is important in MVP development.

## 2. Results

### 2.1. MVP-Affected Valves Have Increased α-SMA Expression

Mitral valves from a total of 37 dogs were used for this study (n = 17 normal valves, n = 20 MVP-affected valves). The demographics of canine patients used in this study are listed in Table 1. Comparing the two groups, age (*p* = 0.001) and body weight (*p* = 0.002) are statistically different, with a higher average age for the MVP-affected group (12.4 ± 2.8 years) than the control group (5 ± 4.7 years). The average weight of the dogs from the MVP-affected group was lower (14.4 ± 9.3 kg) than the control group (22.3 ± 10.6 kg). These differences are expected since MVP is an age-related heart disease and most commonly affects smaller breeds.

We next analyzed the expression of known markers of myofibroblast activation and canonical pathways implicated in this process. Immunohistochemistry of the valve sections showed increased α-SMA expression in valves affected by MVP (Figure 1a), which is most notable in areas next to the abnormal nodular formations with disruption of the normal valve layering. Gene expression analysis of isolated VICs from MVP-affected valves confirmed an increase in *α-SMA* expression, along with increases in *p21*, *TGFβ-receptor 2*, and *TGFβ2* expression (Figure 1b) even when adjusted for age and weight. A statistical analysis looking at the effects of age and weight did not show any differences in these genes between different age and weight groups (Appendix A). Increased α-SMA protein expression was confirmed in VICs isolated from MVP-affected valves by immunoblotting (Figure 1c).

### 2.2. ncRNA Profiles in Both Cells and Extracellular Vesicles Were Altered in MVP

We confirmed vesicles isolated from cultured VICs using sized exclusion chromatography using serum-free media had characteristics of sEV; cup-shaped morphology observed by transmission electron microscopy, particle size distribution centered between 100–200 nm by nanoparticle tracking analysis, and expressions of CD9, TSG101, and Alix by immunoblotting. Immunoblotting also confirmed that the isolated vesicles were negative for calnexin, a protein associated with endoplasmic reticulum, and not sEV (Figure 2a).

The cultured VICs and their associated sEV were sequenced for their small ncRNA contents. Hierarchical clustering and principal component analysis showed clustering of cellular ncRNA profile by disease group (normal vs. MVP) (Figure 2b). Clustering by disease status was also observed in the associated sEV ncRNA profile, although the separation between the two groups was less defined (Figure 2c).

### 2.3. miR-145 Expression Is Upregulated in MVP-Associated VICs and Their sEVs

Sequence analysis showed that two miRNAs were significantly upregulated in both the cells and their associated sEV of diseased valves: *miR-133c* (cell: fold change = 1.6, padj = 0.031; sEV: fold change = 1.7, padj = 0.022) and *miR-145* (cell: fold change = 1.6, padj = 0.029; sEV: fold change = 1.8, padj = 0.019). To validate the sequence results, RT-qPCR was used for cellular RNA and droplet digital PCR was used for sEV RNA. Validation confirmed the upregulation of *miR-145* in MVP-associated VICs and sEV (Figure 3) and the upregulation of *miR-133c* in MVP-associated VICs, but this could not be confirmed in disease-associated sEV. We, therefore, prioritized further investigation of the functional role of *miR-145* and not *miR-133c* in MVP-associated VICs.

### 2.4. miR-145 Inhibits KLF4 That Results in Increased Expression of α-SMA in VICs

To determine the function of *miR-145* in MVP, we modulated *miR-145* expression in VICs by transfection of *miR-145* mimic and *miR-145* inhibitor with Lipofectamine RNAiMAX, followed by transcriptomic analysis of the treated cells (n = 6 cell lines). Hierarchical clustering of the transcriptomic analysis showed distinct separation between the *miR-145* mimic, inhibitor, and control (random miRNA sequence) transcriptome profiles, and principal component analysis showed similarity between the inhibitor and control-treated group profiles and separation from the mimic-treated group profile (Figure 4a). The transcriptomic analysis also showed an increase in *α-SMA* expression (fold change = 53.4, padj = 4 × 10^−11^) with *miR-145* mimic transfection.

We further analyzed the transcriptomic data using Ingenuity Pathway Analysis Upstream Regulator Analysis (Qiagen) to understand which upstream regulators of *α-SMA* are differentially expressed with *miR-145* mimic transfection. Of the list of these upstream regulators, two were downregulated and predicted to be a direct target of *miR-145* in silico (targetscan) [20]: SMAD3 (padj = 7.4 × 10^−8^, fold change = −1.5) and *KLF4* (padj = 0.011, fold change = −1.4) (Figure 4c). As downregulation of SMAD3 is known to have the opposite effect on α-SMA [21], i.e., a decrease in SMAD3 should lead to a decrease in α-SMA instead of the increase that is observed with *miR-145* mimic transfection, we focused on determining the function of KLF4 in VICs as it relates to *miR-145* in the context of VICs in MVP. We further investigated the KLF4 protein expression pattern of VICs harvested directly from canine valves and found that mean KLF4 protein expression was reduced in VICs from MVP-affected valves, although it did not reach statistical significance (*p* = 0.137) (Figure 1c).

Using a luciferase reporter assay to measure the binding of *miR-145* to the 3′ UTR of *KLF4* (Figure 5a), we confirmed the direct inhibition of *KLF4* expression by *miR-145*. Furthermore, we showed that overexpression of KLF4 using recombinant KLF4 protein treatment of VICs leads to a decrease in α-SMA expression (Figure 5b), and *KLF4* siRNA treatment resulted in a decreased protein expression of α-SMA (Figure 5c). Taken together, we showed that overexpression of *miR-145* increases α-SMA expression through inhibition of KLF4 in VICs.

### 2.5. Extracellular Vesicles from Myofibroblastic VICs Can Promote Activation of Quiescent Fibroblastic VICs

Using carboxyfluorescein succinimidyl ester (CFSE) staining of VIC sEV, flow cytometry analysis showed that sEV can be taken up by other VICs when co-cultured in culture media (Figure 6a). Furthermore, co-culture of sEV isolated from conditioned media of myofibroblastic VICs with fibroblastic VICs results in an increased protein expression of α-SMA compared to cells cultured with vehicle (PBS) only (Figure 6b), indicating a phenotypic transition to activated myofibroblasts.

## 3. Discussion

We have demonstrated for the first time that similar miRNA dysregulation occurs in the valve cells and their associated sEV, in myxomatous MVP using the canine model. Furthermore, *miR-145*, found to be overexpressed in both the cells and sEV, plays a functional role in promoting the transition of fibroblast to the myofibroblast phenotype of these mitral VICs. Myxomatous MVP is a polygenic disease in both humans and dogs. Regardless of the genetic causes, the common feature observed in both species is the phenotypic transition of quiescent fibroblastic VICs to myofibroblasts; therefore, if a relevant cell activation pathway that is important in this transition can be found, a treatment target may be designed for all patients even if the inciting genetic cause is unknown. We, therefore, sought to find such a non-coding RNA pathway important in the VIC phenotypic change. Sequence analysis of canine MVP-associated VICs showed that *miR-145* is upregulated in disease-associated cells and in their associated sEV. As MVP is also an aging disease in both species, we cannot distinguish the effects of age on changes in miRNA expression. To determine if *miR-145* upregulation is relevant to the transition from fibroblasts to myofibroblasts, we investigated the functional effects of *miR-145* in VICs. *miR-145* overexpression in cultured VICs recapitulated the cellular phenotype of myofibroblasts (increase in α-SMA expression) found in MVP-affected valves, and *miR-145* inhibitor transfection had the opposite effect on VICs. Transcriptomic sequence analysis and IPA pathway analysis revealed that *miR-145* transfection results not only in an increased *α-SMA* expression but also the downregulation of *KLF4*, a predicted target of *miR-145*. To better understand the interactions between *miR-145*, *KLF4*, and *α-SMA*, direct targeting of *KLF4* by *miR-145* was confirmed with the luciferase assay. We then showed that KLF4 can modulate the expression of α-SMA—specifically, increased KLF4 in VICs inhibited α-SMA expression while downregulation of KLF4 had the opposite effect.

The effect of *miR-145* on α-SMA is not unique to VICs. In vascular smooth muscle cells, overexpression of *miR-145* was found to increase α-SMA expression by direct targeting of either *KLF4* or *KLF5* [22,23,24,25]. In cardiac fibroblasts, *miR-145* increases α-SMA expression through repression of *KLF5*, and inhibiting *miR-145* with antogomir decreases cardiac scar sizes in mice [26]. Similar effects of *miR-145* on α-SMA were noted in lung fibroblasts [27]. Therefore, inhibiting *miR-145* to prevent activation of cells that will ultimately lead to scarring or unwanted morphological changes to tissue structures, such as those seen in mitral valve prolapse, is a potential target for therapy.

Our results also highlight that the disease-relevant changes in cellular ncRNA expressions can be similarly seen in the sEVs. The increase in *miR-145* was found not only in VICs harvested from MVP-affected valves but also in their associated sEV. As sEV can be taken up by neighboring cells for local cell-to-cell communication [28], the content of the sEV produced by myofibroblastic VICs can promote phenotypic transition and activation of neighboring cells and potentially exacerbate the extent of valvular pathology as activated myofibroblastic VICs contributes to the remodeling of the extracellular matrix. This important finding resonates with other recent investigations that demonstrate the importance of local (paracrine) sEV-mediated signaling in cardiac fibrosis [29] or cancer [28]. sEV ncRNA content may also serve as biomarkers for disease progression; however, as our previous work in plasma sEV has shown, attempts to dissect signals only from the mitral valve may be difficult as overall systemic changes such as in heart failure will also contribute to the overall circulating sEV ncRNA content. Work is ongoing to identify tissue-specific sEV markers that can be isolated from plasma, which will increase the specificity of sEV ncRNA signals as a disease biomarker.

### Study Limitations

As our study results were acquired from in vitro experiments, culture conditions (e.g., culture plate stiffness, cell passage) have the potential to influence cell behavior, including miRNA expression. To avoid confounding factors of the culture environment, cells were cultured in a consistent manner, and the changes in miRNA expression between VICs acquired from normal and MVP-affected valves were seen despite these in vitro conditions. Furthermore, age is a confounding factor in our study. However, as MVP is an age-related degenerative disease, the influence of age on *miR-145* expression changes may be important for the development of MVP.

## 4. Materials and Methods

### 4.1. Histologic Characterization of Mitral Valves

The mitral valves were harvested from dogs donated to the Tufts University Cummings School of Veterinary Medicine Foster Hospital for Small Animals under the auspice of the tissue donation program. The protocol for tissue harvest and owner consent has been previously described [14]. Dogs with a history of neoplasia had gross signs of neoplasia, or with congenital heart diseases or other acquired heart diseases other than MVP were also excluded from the study, and at least 1 cm length of mitral valve leaflet needs to be harvested for histological examination. Mitral valves from a total of 37 dogs were harvested and used for this study. These tissues were collected between November 2013 and Jan 2020.

Sections of the mitral valve were fixed in 4% paraformaldehyde for 24 h, then the fixation solution was changed to 70% ethanol. Valves were then embedded in paraffin, sectioned, and stained with hematoxylin and eosin, Sirius red, Mason’s trichrome, pentachrome, and van Gieson’s stains. Classification of valves was performed based on gross examination [9] and histological examination as previously described [14]. Briefly, the collected mitral valves were divided into two groups: normal and diseased. Normal valves were defined as valve leaflets exhibiting no myxoid formation and maintaining the well-organized division of the three main layers within the mitral valve leaflet. If any myxoid formation was observed within the spongiosa or fibrosa layers, the valve leaflet was classified as a diseased valve, and severely affected valves are those with >1 mm^2^/cm of nodular formation.

### 4.2. Mitral Valvular Interstitial Cell (VIC) Isolation and Culture

Valvular interstitial cells were isolated from mitral valves based on a previously established protocol [14]. Briefly, the valve leaflets were first finely minced with a scalpel. The pieces were further digested with a solution consisting of 7 U/mL collagenase I (Sigma–Aldrich, St. Louis, MO, USA), 7 U/mL collagenase XI (Sigma–Aldrich), 4 U/mL DNase (Sigma–Aldrich), and 4 U/mL hyaluronidase (Sigma–Aldrich) in 37 °C with agitation for 1 h. The cells were then filtered through a 70 μm filter and cultured in αMEM media (Lonza, Lexington, MA, USA) with 10% fetal bovine serum (FBS) (Hyclone, Waltham, MA, USA), 0.3 mg/mL L-glutamine (Corning, Corning, NY, USA), and 100 U/mL penicillin and 100 μg/mL streptomycin (Hyclone). After 24 h, the culture media is replaced to remove the unattached valvular endothelial cells. Cells were passaged once (passage 1) and cryopreserved for the subsequent analyses.

VICs (passage 3) were used in this study and first cultured in αMEM media with 10% FBS until 60–70% confluency. Cells were then washed twice with phosphate buffer saline (PBS), and the media was replaced with serum-free defined chemical media [30], composed of DMEM with 25 mM HEPES (Life Technologies, Carlsbad, CA, USA), 1× penicillin–streptomycin, 1× l-glutamine, 1× Insulin-Transferrin-Selenium (Gibco, Waltham, MA, USA), 5 ng/mL recombinant human fibroblast growth factor 2 (Invitrogen, Waltham, MA, USA), and 5 ng/mL recombinant human platelet-derived growth factor AB (Invitrogen). For sEV co-culture experimentation, 10^×9^ particles/mL were added to the serum-free culture media of cultured VICs.

### 4.3. Small Extracellular Vesicle Isolation and Characterization

Cell culture supernatant was collected after VICs (passage 3) were cultured in serum-free defined chemical media for 48 h. The collected media was then centrifuged at 300× *g* for 10 min followed by 2000× *g* for 10 min and a final centrifuge step at 10,000× *g* for 30 min. The supernatant was then filtered through a 0.22 µm syringe filter. sEV were then isolated from the filtered supernatant using size exclusion chromatography (IZON, Medford, MA, USA; qEV10/70 nm) by collection of fractions 5–10. Collected fractions were further concentrated using an Amicon Ultra-15 10 kDa Centrifugal Filter (MilliporeSigma, Burlington, MA, USA).

sEV size distribution and concentration were evaluated with nanoparticle tracking analysis (NTA; NS300, software v3.0; Malvern, Westborough, MA, USA) using the following settings: laser wavelength (488 nm); temperature (23 °C); screen gain (1.0); camera level (13); infusion pump (5 μL/min); five videos recorded per sample (60 s per video); and detection threshold at five with auto blur and auto max jump distance settings. Transmission electron microscopy using uranium acetate negative staining was used to further confirm the presence and visualize morphology of sEV. Western blot was used to confirm the presence of sEV membrane proteins (see Section 4.6).

### 4.4. Gene Overexpression and Inhibition

VICs were cultured as described above in serum-free media. *miR-145* mimic, inhibitor, and negative control miRNA (mirVana, Waltham, MA, USA) were transfected using Lipofectamine RNAiMAX (Thermo Fisher, Waltham, MA, USA) for 48 h. Cells were then harvested for subsequent analyses.

### 4.5. Real-Time PCR and Small RNA Sequencing for Gene Expression Analysis

#### 4.5.1. Real-Time PCR for Cells and sEV

Total RNA, including mRNA and miRNA, was isolated from VICs cultured in serum-free defined chemical media using a mirVana miRNA isolation kit (Invitrogen) following the manufacturer’s instructions. RNA concentration was determined using the 2100 Bioanalyzer System (Agilent, Santa Clara, CA, USA). cDNA for miRNA was synthesized using miRCURY LNA RT Kit (Qiagen, Hilden, Germany) targeting an input of 10 ng of RNA for each sample, and RT-qPCR was performed using miRCURY LNA miRNA PCR Systems (Qiagen) according to the manufacturer’s instructions. *miR-128* was used as the normalization gene for cellular miRNA PCR analysis as it was shown to be the most stably expressed miRNA across all of the samples using NormFinder (v. 20, Aarhus University Hospital, Aarhus, Denmark) [31], and similarly, *miR-125* was used for normalization for sEV miRNA expression analysis. For mRNA, cDNA was synthesized using the RT^2^ HT First Strand Kit (Qiagen) targeting an input of 1000 ng of RNA for each sample. RT-qPCR was performed using RT^2^ SYBR Green qPCR Mastermix (Qiagen), and two housekeeping genes were used (RPS19 and HPRT).

For sEV miRNA isolation, isolation was performed using the miRNeasy Serum/Plasma Kit (Qiagen). cDNA was synthesized with the miRCURY LNA RT Kit, and RT-qPCR was performed using miRCURY LNA miRNA PCR Systems (Qiagen). Droplet digital PCR was performed with the QX600 Droplet Digital PCR System (Bio-Rad, Hercules, CA, USA).

#### 4.5.2. mRNA and Non-Coding RNA Sequencing

For small ncRNA sequencing, total RNA was used as input for sequencing library preparation using QIAseq miRNA Library Kit (Qiagen). Sequence protocol has been previously described [14]. Briefly, the libraries were sequenced on a HiSeq 2500 sequencer (Illumina, San Diego, CA, USA) using High Output V4 chemistry and single read 100 bases format. The raw sequences were processed into demultiplexed files in compressed fastq format (Illumina). Adaptor sequences and unique molecular identifiers were trimmed from the resulting reads using CLC Genomic Workbench v. 11 (Qiagen), and sequences were mapped to miRBase v. 22 [32]. Sequence reads counts normalization and differential expression and principal component analyses were performed using DESeq2 [33].

For mRNA sequencing, total RNA was used as input for sequencing library preparation using TruSeq Stranded mRNA Library Prep (Illumina). The libraries were then sequenced on a HiSeq 2500 sequencer (Illumina) using High Output V4 chemistry. The raw sequence results were processed into demultiplexed files in compressed fastq format (Illumina). Adaptor sequences were trimmed from the resulting reads using CLC Genomic Workbench v. 11 (Qiagen), and sequences were mapped to the canine genome reference CanFam3.1. Sequence reads count normalization and differential expression analysis were performed using DESeq2 [33].

### 4.6. Immunoblotting and Immunohistochemistry for Protein Expression Analysis

Immunohistochemistry for α-SMA (Sigma–Aldrich A5228; monoclonal mouse antibody; 1:500 dilution) and phosphor-ERK (Cell Signaling Technology, Danvers, MA, USA, 9211S, polyclonal rabbit antibody; 1:1000 dilution) were performed on valve sections. Briefly, deparaffinized tissue sections were blocked in 3% BSA for 2 h at room temperature followed by an incubation overnight using the respective primary antibodies or isotype at 4 °C. The tissue sections were then incubated with a fluorescent secondary antibody (Alexa Fluor, Thermo Fisher) for 30 min at room temperature. DAPI (40,6-diamidino-2-phenylindole) was applied to each section.

Protein from cells and EVs for immunoblotting was extracted by incubating with M-PER (Thermo Fischer) for 10 min at room temperature, followed by centrifugation at 14,000× *g* for 15 min. The supernatant was collected and stored at −20 °C prior to protein analysis. Protein concentration was measured by Pierce BCA Protein Assay Kit (Thermo Fisher), and 5–20 µg of total protein was loaded for electrophoresis and transferred to the polyvinylidene difluoride membrane. The membrane was processed for immunoblotting against α-SMA (Sigma–Aldrich A5228; clone 1A4, monoclonal mouse antibody; 1:500 dilution, antibody ID: AB_262054), KLF4 (Abcam, Cambridge, UK, ab129473, clone rabbit polyclonal, 1:1000 antibody ID: AB_2941800), β-actin (Cell Signaling Technology 3700, mouse monoclonal, clone: 8H10D10; 1:1000, antibody ID: AB_2242334), CD9 (Bio-Rad MCA 469, mouse monoclonal, clone: MM2/57; 1:500; antibody ID: AB_323961). TSG101 (BD Biosciences, Franklin Lakes, NJ, USA, 612696, mouse monoclonal, clone: 51/TSG101, 1:1000, antibody ID: AB_399936), Alix (Abcam ab76609, rabbit polyclonal, 1:1000, antibody ID: AB_2042595), and calnexin (Abcam ab75801, rabbit polyclonal, 1:1000, antibody ID: AB_1310022). Secondary antibodies used were anti-rabbit IgG (Vector Laboratories, Newark, CA, USA, BA-1000, goat polyclonal, 1:250, antibody ID: AB_2313606) and anti-mouse IgG (Vector Laboratories BA-2000, horse polyclonal, 1:250, antibody ID: AB_2313581). Immunoblot images for α-SMA and KLF4 were analyzed using ImageJ v. 1.45a [34] and normalized against the β-actin signal.

### 4.7. sEV Visualization and Flow Cytometry

sEVs were first isolated from the 72-h conditioned media of cultured VICs using sequential centrifugation (2000× *g* for 10 min and 10,000× *g* for 30 min, both at room temperature) followed by a final ultracentrifugation step (100,000× *g* for 2 h at 4 °C). The sEV were then incubated with 20 μM of carboxyfluorescein diacetate, succinimidyl ester (CFSE), a dye that binds covalently to intracellular amines, for 2 h at 37 °C. Excess CFSE was removed using a Amicon 10 kDa MWCO filter (MilliporeSigma). Labeling of the sEV was verified with fluorescence signaling using flow cytometry (BD Biosciences Accuri C6 Plus). CFSE-labelled sEV were then incubated with VICs for 16 h, and the cells were washed in PBS prior to characterization.

### 4.8. Luciferase Assay Assessing Binding of miR-145 to KLF4 3′UTR

Luciferase assay was performed to confirm *KLF4* as the target for *miR-145*. Two different binding site sequences, wild-type (5′ ctagc ttacaaaaca ccaaaggggg gtgactggaa ggtgtgaata ttacaaaaca ccaaaggggg gtgactggaa ggtgtgaata g 3′) and mutated (5′ ctagc ttacaaaaca ccaaaggggg gtggtcaaga ggtgtgaata ttacaaaaca ccaaaggggg gtggtcaaga ggtgtgaata g 3′), were inserted into pmirGLO Dual-Luciferase miRNA Target Expression Vector (Promega, Madison, WI, USA). The chemical transformation was performed with the Gibson Assembly Cloning Kit (NE Biolabs, Ipswich, MA, USA). Resultant clone sequences were confirmed by sequencing (wild type and mutated) and subsequently transfected into human embryonic kidney (HEK) cells using Lipofectamine 3000 (Thermo Fisher) along with *miR-145* mimic and negative control (mirVana). inally, luciferase assay was performed with the Dual-Luciferase Reporter Assay (Promega).

### 4.9. KLF4 Stimulation and siRNA Inhibition of Valvular Interstitial Cells

Human recombinant KLF4 protein (Abcam ab169841) was incubated with protein transfection reagent (Pierce, Thermo Fisher) following the manufacturer’s instruction. The KLF4 protein (0.1 ng/mL) was then added to VICs cultured in serum-free media for 24 h. Cells were then harvested for subsequent analyses.

*KLF4* siRNA inhibition was performed using a combination of three Dicer-substrate siRNAs (IDT, Coralville, IA, USA). The siRNA sequences were designed to target canine *KLF4* cDNA sequences. The sequences were (1) 5′-gccaacuugugaguggauaau-3′; (2) 5′-gguguggauaucaggguauaaauta3′; and (3) 5′-cuagaaagcacuacaaucaugguca -3′. Non-targeting negative control siRNA was used as negative control (Invitrogen Silence Select Negative Control No. 1), and a vehicle-only added condition was used as control. Transfection of siRNA was performed with Lipofectamine RNAiMAX.

### 4.10. Statistical Analysis

Statistical analysis was performed using commercially available software (GraphPad Prism version 10.0.0, (Dotmatics, Boston, MA, USA). Data normality was analyzed with the Shapiro–Wilk test. Normally distributed data was analyzed with a One-way analysis of variance (ANOVA) test for multiple group comparison and a t-test for two-way comparison, and the Kruskal–Wallis test was used for non-parametric statistical testing for multiple group comparison and the Mann–Whitney test for two-way comparisons.

## Figures and Tables

**Figure 1 ijms-25-01468-f001:**
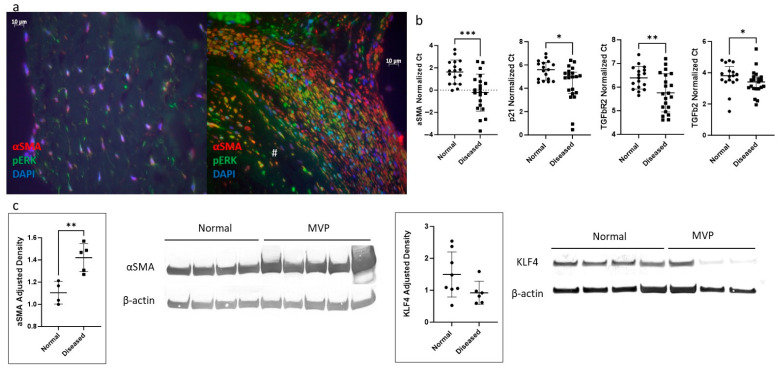
Canine VIC associated with MVP has myofibroblastic phenotype. (**a**) Representative immunohistochemistry images showing increased α-SMA (red) seen in MVP-affected valve (**right**) with # noting a myxoid structure compared to normal valve (**left**). DAPI (blue), phosphor-ERK (green). (**b**) Gene expression was measured by PCR showing VICs harvested from MVP/disease valves with lower Ct number (higher expression) of *α-SMA*, *p21*, *TGFβR2*, and *TGFβ*. n = 20 for disease/MVP group, and 17 for normal group. (**c**) Immunoblotting confirms increases in protein expression α-SMA in VICs harvested from MVP valves. n = 5 for disease/MVP group, and 4 for normal group. KLF4 protein expression normalized to β-actin was lower in diseased group but did not reach statistical significance. n = 3 for disease/MVP group, and 4 for normal group. •: normal VIC cell lines; ▪: MVP VIC cell lines. *: *p* < 0.05; **: *p* < 0.01; ***: *p* < 0.001. For normally distributed data: unpaired *t*-test, mean, and standard deviation shown; for non-normally distributed data: Mann–Whitney test, median, and interquartile range shown.

**Figure 2 ijms-25-01468-f002:**
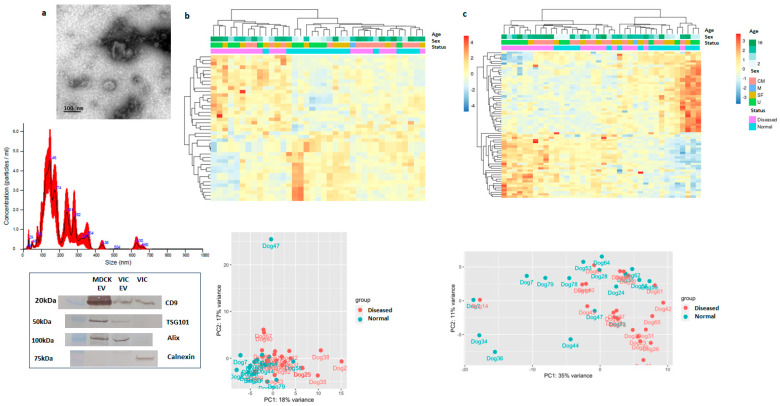
Cellular and sEV ncRNA profiles differ between normal and MVP-affected valves. (**a**) Small EVs collected from VICs showed expected cupped-shaped morphology under transmission electron microscopy (**top**), size distribution between 100–200 nm measured by nanoparticle tracking analysis (**middle**), and protein expressions positive for CD9, TSG101, Alix, and negative for calnexin as measured by immunoblotting (**bottom**). MDCK sEVs served as positive sEV protein control (CD9, TSG101, Alix), and VIC cellular protein served as positive control for cellular protein (calnexin). (**b**) Hierarchical clustering analysis (**top**) of cellular small ncRNA expression was analyzed against age and sex of the study dogs, showing clustering by disease group (normal vs. diseased). Each column represent individual study dogs, and each row represent a non-coding RNA species with padj < 0.05 Principal component analysis demonstrate the same clustering by disease group (**bottom**). (**c**) Hierarchical clustering (**top**) and principal component analysis (**bottom**) of sEV ncRNA.

**Figure 3 ijms-25-01468-f003:**
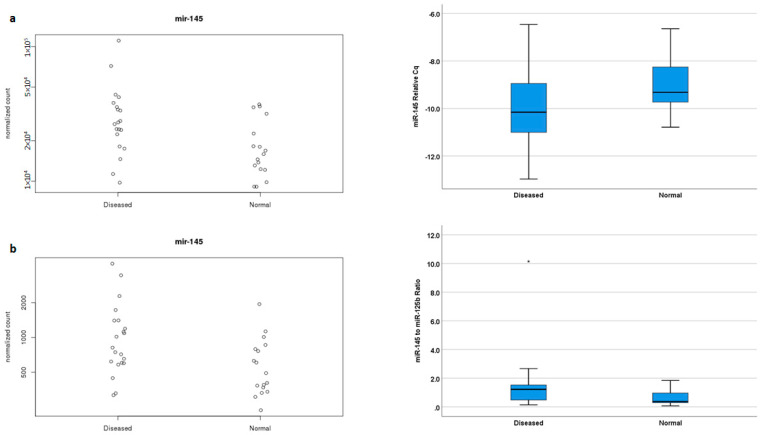
miR-145 is upregulated in VICs and sEVs associated with diseased valves. (**a**) Normalized sequence counts of *miR-145* (**left**) are increased in VICs harvested from MVP valves. This upregulated expression was confirmed secondarily by RT-qPCR (**right**) showing lower Cq (Ct) number for diseased VIC. n = 20 for diseased group; 17 for normal group; *p* = 0.007; unpaired *t*-test, mean and standard deviation shown. (**b**) *miR-145* expression was also upregulated in the associated sEVs as demonstrated by the normalized sequence count (**left**), which was secondarily confirmed by droplet digital PCR (**right**), showing an increase in copy number ratio between *miR-145* to *miR-125* n = 20 for diseased group; 17 for normal group; *p* = 0.032; Mann–Whitney test, median, and interquartile range shown. * represents an outlier.

**Figure 4 ijms-25-01468-f004:**
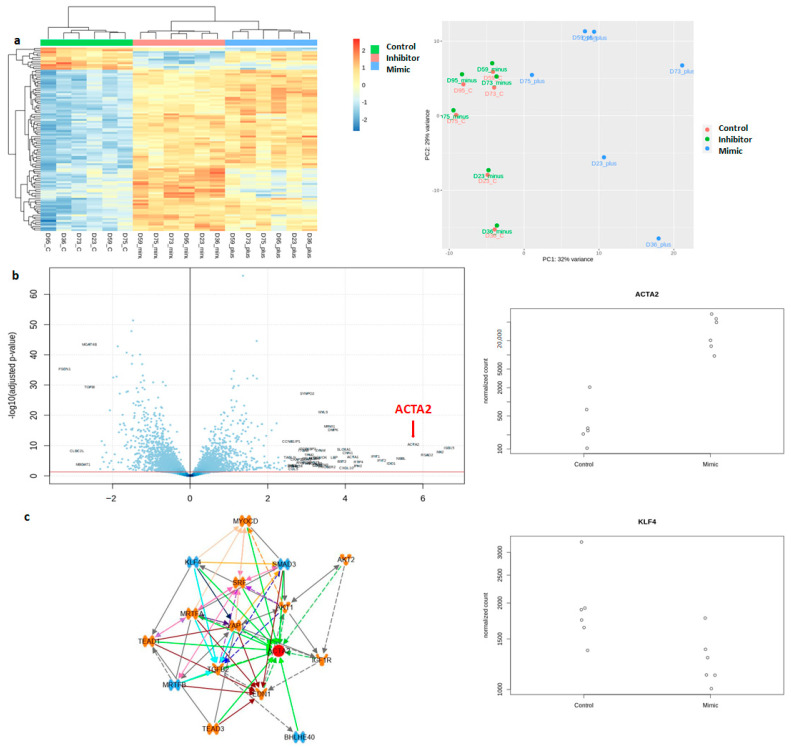
*miR-145* upregulation results in increased *α-SMA* expression and decreased *KLF4* expression. (**a**) Transfection of *miR-145* mimic, inhibitor, and control miRNA into VICs (n = 6/group) showed distinct transcriptomic profiles on hierarchical clustering (**left**) and separation between groups on principal component analysis (**right**). The *miR-145* inhibitor transfected cells have similar profiles as control miRNA transfected cells, while *miR-145* mimic transfected profile is distinct from the other two groups. (**b**) Volcano plot (**left**) showed that miR-145 mimic transfected cells resulted in an upregulation of *α-SMA* (ACTA2, red arrow pointing at the data point for ACTA2) compared to control sample (padj = 4 × 10^−11^) (**middle**). (**c**) Ingenuity Pathway Analysis Upstream Regulator Analysis (**left**) identified 11 upregulated genes (orange) and four that are downregulated genes (blue) that could regulate *α-SMA* expression after *miR-145* mimic transfection. *KLF4* is one of the genes with decreased normalized count after mimic transfection compared to control (padj = 0.011) (**right**).

**Figure 5 ijms-25-01468-f005:**
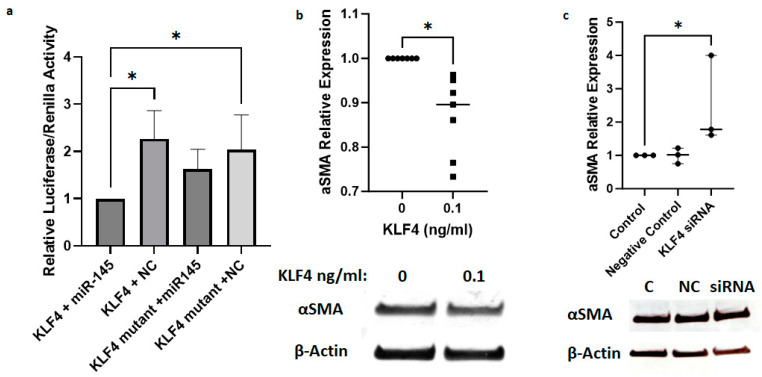
*miR-145* directly targets *KLF4* to suppress *KLF4* expression. (**a**) Luciferase assay confirms targeting of *KLF4* by *miR-145*. Assay was performed using HEK cells with control miRNA (NC). n = 3 wild-type and 3 mutant vectors; one-way ANOVA, mean, and standard deviation shown. (**b**) Immunoblot confirmed that VIC co-culture with KLF4 protein decreases α-SMA protein expression (n = 7/group; paired *t*-test, mean, and standard deviation shown), and (**c**) *KLF4* siRNA treatment resulted in an increase in α-SMA protein expression. C: Control (vehicle only), NC: Negative control siRNA, siRNA: *KLF4* siRNA (n = 3/group; Kruskal–Wallis test, median, and interquartile shown). *: *p* < 0.05.

**Figure 6 ijms-25-01468-f006:**
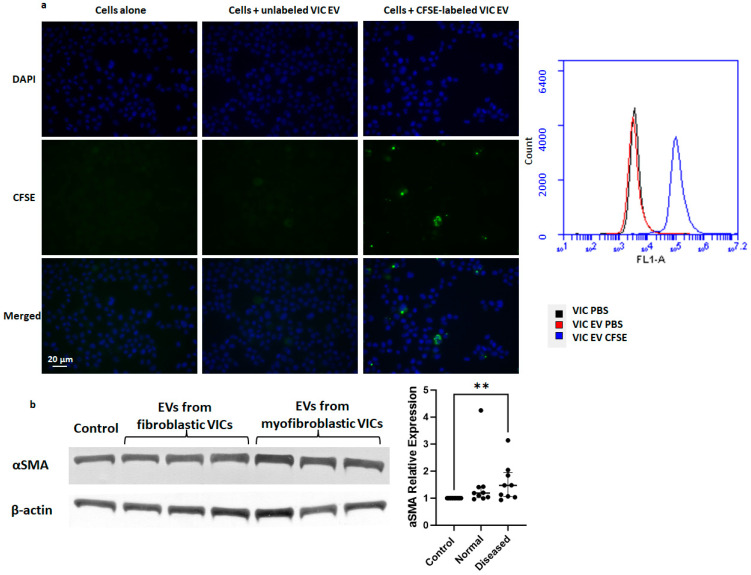
sEV with dysregulated miRNA are taken up by quiescent VICs, resulting in phenotypic transition to myofibroblasts. (**a**) CSFE staining of sEV isolated from conditioned media of fibroblastic VICs is shown to be taken up by VICs during co-culture with colocalization of CSFE and DAPI signals under fluorescent microscopy, and an increase in CSFE intensity with co-culture was noted under flow cytometry. Green: CSFE; Blue: DAPI. (**b**) Immunoblot confirmed that fibroblastic VICs isolated from normal valves co-culture with sEV produced by myofibroblastic VICs isolated from diseased/MVP valves resulted in an increase in α-SMA protein expression compared to vehicle/PBS control. **: *p*< 0.01, n = 9/group, Kruskal–Wallis test, median, and interquartile shown.

**Table 1 ijms-25-01468-t001:** Patient demographics.

Normal Valves:					
Breed	Age (Year)	Sex	Body Weight (kg)	Clinical Findings	Histological Findings
Beagle	4	U	15	No murmur	Normal
German Shepherd	2	U	30	No murmur	Normal
German Shepherd	2	U	30	No murmur	Normal
Doberman Pincher	10	U	30	No murmur	Normal
Belgian Malinois	2	SF	25	No murmur	Normal
Belgian Malinois	4	U	25	No murmur	Normal
Mixed	15	SF	20	No murmur	Normal
Papillon	15	CM	8	No murmur	Normal
Labrador Retriever	7	CM	39.6	No murmur	Normal
Mixed	2	SF	26.4	No murmur	Normal
Mixed	13	CM	27.8	No murmur	Normal
Mixed	7	CM	30	No murmur	Normal
Mixed	11	SF	9.8	No murmur	Normal
Schnauzer	12	F	7.7	No murmur	Normal
Mixed	5	CM	25.7	No murmur	Normal
Chihuahua	5	SF	4.9	No murmur	Normal
Mixed	4	F	4.9	No murmur	Normal
MVP Valves:					
Beagle	9	CM	22.1	No murmur	Severe
Chihuahua	5	SF	4.9		
German Shepherd	9	U	30	No murmur	Severe
Mixed	15	U	10	No murmur	Severe
English Bulldog	7	U	25	No murmur	Severe
Mixed	11	U	5	High-grade murmur	Severe
Bichon Frise	16	U	10	Mid-grade murmur	Severe
Chihuahua	13	SF	4.8	High-grade murmur	Severe
Pomeranian	7	CM	4.5	High-grade murmur	Severe
Cavalier King Charles Spaniel	13	SF	4.5	High-grade murmur	Severe
Chihuahua	12	SF	4	Unknown	Severe
Miniature Pinscher	14	CM	4.5	Mid-grade murmur	Severe
Golden Retriever	11	M	32	Unknown	Severe
Shih Tzu	15	CM	7	Unknown	Severe
Australian Shepherd	14	SF	19.1	Unknown	Severe
Mixed	14	CM	15	Unknown	Severe
Mixed	11	CM	5.8	Mid-grade murmur	Severe
Labrador Retriever	15	CM	24.5	Unknown	Severe
Cavalier King Charles Spaniel	11	F	13.6	High-grade murmur	Severe
Whippet	16	CM	18.2	Mid-grade murmur	Severe

U: Unknown, SF: spayed female; CM: castrated male; F: intact female; Normal: no myxoid nodule seen; Severe: >1 mm^2^/cm of myxoid nodular structure measured in valve leaflet.

## Data Availability

All data presented in this article can be found in Dryad (https://doi.org/10.5061/dryad.dz08kps4h (assessed on 16 January 2024)).

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
