# Peer review of "Defining the Role of the miR-145—KLF4—αSMA Axis in Mitral Valvular Interstitial Cell Activation in Myxomatous Mitral Valve Prolapse Using the Canine Model"

_ijms, 2024, doi:10.3390/ijms25031468_

Round 1

Reviewer 1 Report

Comments and Suggestions for Authors

The Matherial and Methods section is placed after Study Limitation, which is unusual and out of order. 

There is no connections with any clinical data in the manuscript, for example with the level of MVP before harvesting or sighns of cardiac insuficinecy Those relationship will be important to bring those results closer to the clinicians. The role of mi R145 in VIC and seV in myofibroblast activation is very important as well as cellullar nc RNA expression, not only for therapeutic, but also in the future as a part of stratification model. 

Comments on the Quality of English Language

Quality of English Language is very good. 

Author Response

We thank the reviewers for their suggestions and have incorporated changes based on their recommendation.  Here are our responds to the specific questions that the reviewers have listed:

Reviewer 1:

  1. The Material and Methods section is placed after Study Limitation, which is unusual and out of order.

Response:  We have moved the Materials and Methods section in front of the Results section as suggested by the reviewer.

  1. There is no connections with any clinical data in the manuscript, for example with the level of MVP before harvesting or signs of cardiac insufficiency Those relationship will be important to bring those results closer to the clinicians. The role of mi R145 in VIC and sEV in myofibroblast activation is very important as well as cellullar nc RNA expression, not only for therapeutic, but also in the future as a part of stratification model.

Response:  We thank the reviewer for pointing out the importance of the clinical status of the study dogs.  We have updated Table 1 to include any relevant clinical findings available through the patients’ medical records and our histological assessment of the mitral valve leaflets.

Reviewer 2 Report

Comments and Suggestions for Authors

Dear Authors,

Thank you for the opportunity to read a nice article. Overall, it's a well-written article, so I'd like to thank the authors. Although the frequency of MVP in the general population is not clearly known, it is assumed to vary between 2-8%. In this study, it seems that a good result was obtained due to reasons such as the use of a dog model and similarity. Although there is not much room to criticize the article in general terms;

1- The material method part must be placed before the result.

2- The many abbreviations used in the study should be reduced

3- The start and end dates of the study must be specified.

4- The acceptance and rejection criteria in the study must be clearly stated.

5- Its contribution to the literature should be explained in a few sentences.

Kind regards.

Comments on the Quality of English Language

Minor editing of English language required

Author Response

We thank the reviewers for their suggestions and have incorporated changes based on their recommendation.  Here are our responds to the specific questions that the reviewers have listed:

Reviewer 2:

  1. The material method part must be placed before the result.

Response:  We thank the reviewer for suggesting how to improve the manuscript and have moved the Materials section before the Results section.

  1. The many abbreviations used in the study should be reduced

Response:  We have removed 3 abbreviations with 3 remaining, and we hope this makes the manuscript easier to read.

  1. The start and end dates of the study must be specified.

Response:  We have added the start and end dates for sample collection into the methods section (lines 90-92).

  1. The acceptance and rejection criteria in the study must be clearly stated.

Response:  We have added additional inclusion/exclusion criteria into the methods section (lines 90-92).

  1. Its contribution to the literature should be explained in a few sentences.

Response:  We thank the reviewer for suggesting adding a section on our contribution to literature.  This has been added into the discussion section (Line 413).

Reviewer 3 Report

Comments and Suggestions for Authors

General comments

This is a manuscript addressing a topic “Defining the role of the miR-145 – KLF4 – αSMA axis in mitral valvular interstitial cell activation in myxomatous mitral valve prolapse using the canine model”. However, some concerns need to be addressed.

Specific comments

1)      Limitation: no age-matching; The authors mentioned about age in the limitation (Line 276) and discussion (Line 223) and also stated that the gene expression of αSMA was not dependent on the age (Line 95). An additional graph would be informative indicating αSMA as a function of age.

2)      Figure 3 is absent.

3)      In the method 4.5.1, the internal control or the amount of total RNA were not mentioned. Only the comparison with Ct numbers would not be correct.

Author Response

We thank the reviewers for their suggestions and have incorporated changes based on their recommendation.  Here are our responds to the specific questions that the reviewers have listed:

Reviewer 3:

  1. Limitation: no age-matching; The authors mentioned about age in the limitation (Line 276) and discussion (Line 223) and also stated that the gene expression of αSMA was not dependent on the age (Line 95). An additional graph would be informative indicating αSMA as a function of age.

Response: We thank the reviewer for their suggestion.  A graph comparing the normalized aSMA Ct number (from PCR) has been added as Supplemental Figure 1.

  1. Figure 3 is absent.

Response:  We apologize for the omission and have made sure that Figure 3 is added (Line 332)

  1. In the method 4.5.1, the internal control or the amount of total RNA were not mentioned. Only the comparison with Ct numbers would not be correct.

Response: We have clarified the methods for internal control (amount of RNA used for cDNA synthesis and housekeeping genes used) in lines 153-160.

Round 2

Reviewer 2 Report

Comments and Suggestions for Authors

Dear Authors,

I reviewed the article. It seems that the authors did what the referees wanted. There is no obstacle to accepting the article in its current form. I would like to thank the authors and editor.

Kind regards.

Reviewer 3 Report

Comments and Suggestions for Authors

The authors have corrected the maniscript according to the reviewer's comments.